# Factors Associated with Long COVID-19 in a French Multicentric Prospective Cohort Study

**DOI:** 10.3390/ijerph20176678

**Published:** 2023-08-29

**Authors:** Nagham Khanafer, Laetitia Henaff, Sabrina Bennia, Anne Termoz, Roland Chapurlat, Vanessa Escuret, Mathilde Proriol, Florence Duvert, Camille Mena, Catherine Planckaert, Nadège Trehet-Mandez, Mitra Saadatian-Elahi, Philippe Vanhems

**Affiliations:** 1Centre International de Recherche en Infectiologie (CIRI), Inserm U1111, CNRS UMR5308, ENS de Lyon, Lyon 1 University, CEDEX 07, 69364 Lyon, France; laetitia.henaff@chu-lyon.fr (L.H.); mitra.elahi@chu-lyon.fr (M.S.-E.); philippe.vanhems@chu-lyon.fr (P.V.); 2Department of Hygiene, Epidemiology, and Prevention, Edouard Herriot Hospital, Hospices Civils de Lyon, 69003 Lyon, France; sabrina.bennia@chu-lyon.fr; 3Service Recherche et Epidémiologie Cliniques, Hospices Civils de Lyon, Pôle Santé Publique, 69003 Lyon, France; anne.termoz@chu-lyon.fr; 4Department of Rheumatology, Edouard Herriot University Hospital, Hospices Civils de Lyon, 69003 Lyon, France; roland.chapurlat@chu-lyon.fr (R.C.); mathilde.proriol@chu-lyon.fr (M.P.); 5INSERM UMR 1033, University of Lyon, 69003 Lyon, France; florence.duvert@chu-lyon.fr (F.D.); ext-camille.mena@chu-lyon.fr (C.M.); ext-c.planckaert@chu-lyon.fr (C.P.); ext-n.trehet-mandez@chu-lyon.fr (N.T.-M.); 6Prévention des Maladies Osseuses, Edouard Herriot Hospital, Hospices Civils de Lyon, 69003 Lyon, France; 7Institut des Agents Infectieux, Hospices Civils de Lyon, 69317 Lyon, France; vanessa.escuret@chu-lyon.fr

**Keywords:** COVID-19 pandemic, SARS-CoV-2, long COVID-19, post-COVID-19 syndrome, risk factors

## Abstract

(1) Background: A substantial proportion of COVID-19 patients continue to experience long-lasting effects that hamper their quality of life. The objectives of this study were (1) to report the prevalence of persistent clinical symptoms 6–12 months after the onset of COVID-19 and (2) to identify potential factors at admission associated with the occurrence of long COVID. (2) Methods: A prospective study was conducted among COVID-19 adult patients, hospitalized in four French university hospitals. Patients were invited to two ambulatory follow-up medical visits, 6–8 months (visit #1) and one year (visit #2) after the onset of their COVID-19. A multivariate logistic regression was performed to assess factors associated with long COVID. (3) Results: In total, 189 patients participated in this study (mean age of 63.4 years). BMI > 30 kg/m^2^ (aOR 3.52), AST levels between 31 and 42 U/L (aOR 8.68), and AST levels > 42 U/L (aOR 3.69) were associated with persistent clinical symptoms at visit #1. Anosmia (aOR 13.34), AST levels between 31 and 42 U/L (aOR 10.27), stay in ICU (aOR 5.43), pain (aOR 4.31), and longer time before hospitalization (aOR 1.14) were significantly associated with persistent clinical symptoms at visit #2. Patients with ageusia (aOR 0.17) had a lower risk of long COVID. (4) Conclusions: This study showed that some patients experienced persistent clinical symptoms one year after COVID-19 onset that were associated with some determinants at the acute phase/stage.

## 1. Introduction

Coronavirus disease 2019 (COVID-19) is caused by severe acute respiratory syndrome coronavirus 2 (SARS-CoV-2) and can manifest a wide spectrum of disease severity, from asymptomatic to fatal forms [1]. More than two years after the start of COVID-19, a growing proportion of the population who have recovered from acute infection have suffered from persistent symptomatology that has lasted from weeks to years. This status is defined as ”long COVID” or ”post-acute sequelae of SARS-CoV-2 infection” [2]. Long COVID consists of a set of symptoms related to the respiratory, cardiovascular, neurological, endocrine, urinary, and immune systems. Patients suffering from long COVID display persistent or relapsing or new symptoms that include fatigue, dyspnea, cough, myalgia, cardiac or skin anomalies, sleep trouble, post-traumatic stress disorder, and cognitive impairment [3].

The risk factors and the extent of the severity of clinical symptoms of long COVID are subject to debate. The persistence and severity of long COVID appear to be associated with the severity at the acute phase onset (such as the need for mechanical ventilation, admission to the intensive care unit (ICU), the presence of comorbidities, older age, and others. However, these findings have not been confirmed in different populations [1,4,5]. This may be explained by the heterogeneity of the included populations, collected data, variables included in multivariate models, severity of illness, and the follow-up period [6,7,8]. There have been many attempts to organize and systematize post-COVID symptoms. In a recent systematic review and meta-analysis, more than 50 symptoms were reported to be associated with sequelae of COVID-19. Furthermore, 80% of patients with COVID-19 reported having one or more symptoms for more than two and up to 16 weeks after recovery [9]. A large cross-sectional population-based cohort in France found that persistent physical symptoms, 10–12 months after the first wave of the COVID-19 pandemic, were more associated with a patient’s belief of having SARS-CoV-2 infection rather than laboratory-confirmed COVID-19 [10]. The objectives of this study were (1) to report the prevalence of persistent clinical symptoms 6–12 months after the first episode of COVID-19 and (2) to identify potential factors at admission associated with the occurrence of long COVID in patients hospitalized in four French university hospitals.

## 2. Materials and Methods

### 2.1. Study Design

A prospective study was conducted among laboratory-confirmed COVID-19 patients hospitalized in four university-affiliated hospitals (Hospices Civils de Lyon, Lyon, France) to evaluate the prevalence of persistent clinical symptoms 6–12 months after the first episode of COVID-19. Then, a case-control study was completed to evaluate the factors associated with long COVID. Cases were patients who presented at least one persistent clinical symptom, and controls were asymptomatic patients.

### 2.2. Study Population

Laboratory-confirmed COVID-19 patients who had been previously enrolled in the NOSO-COR cohort study [11] were eligible to participate. This study was approved by the National Ethical Committee (Comité de Protection des Personnes, Ile de France V, 14 October 2020), registered in ClinicalTrials (NCT04637867), and conducted in accordance with Good Clinical Practices of the European General Data Protection Regulation dated as of 25 May 2018 (“GDPR”), (collectively, “Data Protection Legislation”).

### 2.3. Patients and Follow-up Visits

After the exclusion of deceased NOSO-COR patients and those who had several hospitalizations (to reduce confounding issues, such as disability related to other illnesses), the remaining 762 patients who had been previously hospitalized in one of the four Lyon University affiliated hospitals were informed about the study by a letter sent to their domicile. A total of 189 patients agreed to participate in the study. These patients were invited to two ambulatory follow-ups, at 6–8 months and at one year after the initial COVID-19 onset.

Written consent was obtained at the first medical visit. At each visit, the patients were interviewed by the study physician about persistent or newly occurring symptoms, any medical event during the window time, close contact with confirmed COVID-19 patients, and influenza and/or COVID-19 vaccination since their initial hospital discharge. Blood, nasopharyngeal, and sputum samples were collected at each visit.

Data regarding the initial episode were previously described in the NOSO-COR cohort study [11].

### 2.4. Definitions

Long COVID was defined as the presence of at least one persisting clinical symptom of COVID-19 since hospital discharge [12]. We did not investigate cognitive and psychological disorders.

### 2.5. Outcomes

The primary outcome was to estimate the prevalence of long COVID, defined as the persistence of at least one clinical symptom (i.e., fatigue, dyspnea, cough, headache, pain, anosmia, ageusia, diarrhea), six months to one year after COVID-19 onset.

The secondary endpoint was the identification of potential factors associated with long COVID based on the clinical, laboratorial, and X-ray features of their first episode of acute COVID-19.

### 2.6. Statistical Analysis

A descriptive analysis was performed, and the frequency of long COVID symptoms was determined. Continuous variables are summarized by mean (min–max) and categorical variables are described by frequency and percentage. A Shapiro–Wilk test was used to assess the normality of data. The comparison of categorical variables was performed by using Fisher or chi-square tests and Student or Mann–Whitney tests for continuous variables. A multivariate logistic regression analysis with stepwise forward selection was performed to assess factors associated with long COVID. A *p*-value of ≤0.20 in the univariate analysis was defined as the criteria for model inclusion. At the second visit, univariate and multivariate analyses were limited to patients who suffered from persistent clinical symptoms at the first and second visits. These patients were compared to asymptomatic patients for both visits. A stepwise regression was designed to find the most parsimonious set of predictors that are most effective in predicting the dependent variable (i.e., long COVID, yes or no according to our definition). Variables are added to the logistic regression equation one at a time, using the statistical criterion of reducing the likelihood error for the included variables. The Hosmer–Lemeshow goodness-of-fit test was used to assess the model-to-data fit. The results were expressed as adjusted odds ratios (aORs) and their 95% confidence intervals (CI 95%). All *p*-values were two-tailed and *p* < 0.05 was considered to be statistically significant. Statistical analyses were performed using IBM SPSS Statistics for Windows, Version 21.0. IBM Corp: Armonk, NY, USA.

## 3. Results

### 3.1. Patient Characteristics

Between November 2020 and May 2021, 189 COVID-19 patients, who had been hospitalized during the first wave of the pandemic in France (26 February to 19 June 2020), agreed to participate in this study. The demographic characteristics of these patients were similar to those informed about this study. Table 1 summarizes the demographic and clinical characteristics of the study population at hospital admission. The mean age was 63.4 years (min–max, 25–93) and 51% of the participants were over 65 years old. Males represented 59.8% of the study participants, and 57% of the participants had never smoked. The mean body mass index (BMI) was 27.4 kg/m2 (16.4–44.4). In total, 151 patients (79.9%) had underlying comorbidities, mainly cardiovascular diseases (44.4%), hypertension (36%), and diabetes (19%). The mean number of symptoms during the acute COVID-19, was 5 (min–max, 1–9) with a mean duration of 24.5 days (min–max, 2–1260; they included cough (79.8%), fatigue (73.9%), shortness of breath (66%), pain (37.8%), and diarrhea (33.5%). Among the 189 participants, 64 (34.0%) participants required hospitalization in intensive care units (ICU) with a mean length of stay of 20.0 days (min–max, 1–1840. The biological results were in the mean range except for aspartate transaminase (AST), C-reactive protein (CRP), lactate dehydrogenase (LDH), and urea levels.

### 3.2. Prevalence of Clinical Symptoms at Follow-up Visits

The prevalence of persistent clinical symptoms at follow-up visits are summarized in Table 2. Among the 178 patients who completed the two follow-up visits, 86 (48.3%) patients had reported persistent clinical symptoms since the beginning of their infection. The prevalence of clinical symptoms at the acute phase, i.e., visit #1, and visit #2, are shown in Figure 1.

Overall, 109 (57.7%) and 103 (57.9%) patients declared the persistence of at least one symptom at visit #1 and visit #2, respectively. New clinical symptoms were reported by 27 (14.3%) patients at visit #1 and 9 (5.1%) patients at visit #2.

The most persistent clinical symptoms at visit #1 (mean 8.8 months) were dyspnea (37.6%), fatigue (33.9%), and pain (23.9%). The mean number of symptoms decreased significantly from five symptoms (min–max, 1–9) at the onset to 1.5 (min–max, 0–10) at visit #1 (*p* < 0.001). The scatter plot does not depict a correlation between the number of clinical symptoms at the onset of COVID-19 infection and those self-reported at visit #1 (*p* = 0.10).

The most reported symptoms at visit #2 (mean 12 months) were dyspnea (24.7%) and pain (24.2%). New-onset clinical symptoms, mainly fatigue, were reported in nine patients (5.1%). The number of persistent clinical symptoms was significantly lower than described at the onset of COVID-19 (*p* < 0.001); they showed no correlation (*p* = 0.21).

### 3.3. Factors Associated with Persistent Clinical Symptoms at Follow-up

Table 3 and Table 4 summarize the results of the univariate and multivariate regression analyses. The univariate logistic regression showed that a body mass index (BMI) >30 kg/m^2^, ICU admission or complications during initial hospitalization, duration of initial symptoms, and aspartate transaminase (AST) levels between 31 and 42 U/L were significantly associated with persistent clinical symptoms at the two visits.

In the multivariate analysis, BMI > 30 kg/m^2^ (aOR 3.52, 95% CI 1.25–9.91), AST level between 31 and 42 U/L (aOR 8.68, 95% CI 2.41–31.28), and AST level >42 U/L (aOR 3.69, 95% CI 1.32–10.31) were associated with persistent clinical symptoms at visit #1.

Anosmia (aOR 13.34, 95% CI 2.07–86.21), AST levels between 31 and 42 U/L (aOR 10.27, 95% CI 1.88–56.29), stay in ICU (aOR 5.43, 95% CI 1.39–21.25), pain (aOR 4.31, 95% CI 1.23–15.05), and longer time between the onset of clinical symptoms and hospitalization (aOR 1.14, 95% CI 1.01–1.29) were independently associated with persistent symptoms 12 months after acute COVID-19. While patients who suffered from ageusia (aOR 0.17, 95% CI 0.03–0.92) had a lower risk of long COVID, patients with rheumatic diseases (aOR 5.25, 95% CI 0.75–36.84) or a BMI > 30 kg/m^2^ (aOR 3.80, 95% CI 0.97–14.91) showed a higher risk of long COVID; the latter association was not statistically significant.

## 4. Discussion

After recovery from acute COVID-19, some patients continue to experience symptoms and disabilities weeks to months after the initial episode of COVD-19; a diagnosis of long COVID syndrome should be considered if other alternative diagnoses have been excluded [13]. The pathophysiological mechanisms underlying post-acute COVID-19 symptoms are likely to be multifactorial. In this study, long COVID was reported in 48% of patients one year after the initial hospitalization.

Previous studies on the prevalence of long COVID, summarized in a systematic review, found that the median proportion of patients with at least one persistent clinical symptom ≥60 days after diagnosis or at least 30 days after recovery was 72.5% [7]; 50–70% hospitalized patients did not fully recover [14]. The prevalence of persistent COVID-19 was high in the herein study, as well as in the previous systematic review. Lower prevalence of persistent clinical symptoms has been reported in a community-based study in the Netherlands [15]. The estimation of prevalence depends on the length of follow-up, the included population, and the definition of long COVID, which can explain the heterogenic results in the literature [7].

The most reported long COVID symptoms during both visits were pain and dyspnea, while fatigue was mainly reported during the first visit. Other studies have reported similar rates and patterns of long COVID symptoms [6,16,17]. In the WHO Delphi procedure, fatigue and dyspnea were considered to be the most important symptoms of the post-COVID-19 condition [12]; however, in our patients, other distinctive symptoms were reported, including pain (mainly myalgia), ageusia, or anosmia. These symptoms were recently described in a study of the general population [15].

Our results showed that ICU admission was significantly associated with an increased risk of long COVID. Previous studies have described a high burden of persistent clinical symptoms in patients admitted to the ICU for severe and critical COVID-19, but the follow-up was only 2 to 6 months following ICU discharge [18,19,20]. Before the pandemic, post-intensive care syndrome was described in patients admitted to the ICU [21], which highlighted the need to find a management plan through both pharmacologic and non-pharmacologic modalities to be used for this group of patients.

Our results showed consistency with previous reports that anosmia in the acute phase was associated with a higher risk of long COVID. Respiratory viruses are known to cause qualitative olfactory disorders [22,23,24]. The incidence of olfactory disturbances after COVID-19 is very high and its persistence increases patient anxiety and reduces quality of life [25]. In addition to olfactory impairment, the detection of both SARS-CoV-2 RNA and protein in brain tissue is considered to be an important diagnostic and prognostic indicator for long-term neurological complications associated with COVID-19 [26].

Our study showed that AST levels between 31 and 42 U/L are independently associated with long COVID. Abnormal liver function and liver injury related to COVID-19 during hospitalization have been reported, and they can persist as long as one year after hospital discharge. [27]. Alterations in liver function are associated with fatigue and muscle pain, prompting the screening of liver function tests on admission and during the follow-up period.

At admission, 38% of patients reported pain (muscle pain, joint pain, chest pain, or abdominal pain). The multivariate analysis showed that this symptom was associated with long COVID one year after the initial episode of COVID-19. Indeed, fatigue is thought to be a long-lasting symptom of post-COVID with a slow recovery curve during the following years after the infection, regardless of the SARS-CoV-2 variant [28]. There is increasing evidence that individuals with long COVID share common symptoms with myalgic encephalomyelitis/chronic fatigue syndrome. A better understanding of the mechanisms behind post-COVID-19 fatigue is needed to improve the management of patients suffering from long COVID [29].

The main strength of this study was its prospective design in which the data were collected starting from the first hospital admission in addition to a scheduled assessment of persistent or newly occurring clinical symptoms after hospital discharge. Furthermore, the included patients were followed as long as 12 months after the acute COVID-19. Currently, the most frequently used description for long COVID refers to symptoms that last more than three months after the onset [30]. RT-PCR tests were performed for both visits, which permitted us to decrease the likelihood that persistent clinical symptoms could be related to a new episode of COVID-19. Finally, face-to-face interviews with physicians reduced information bias and allowed an accurate data collection.

However, the relatively small sample size limited the statistical power for detecting differences, and the study population was limited to those previously hospitalized in Lyon University hospitals, which biased the representativeness of our sample. Second, the symptoms we studied herein have already been reported in previous reports on long COVID. Third, the prevalence of persistent clinical symptoms was based on what was subjectively reported by the included patients, which could have been influenced by psychological and cognitive factors, especially in the elderly. In fact, cognitive and psychological disorders were not investigated in this study. Davies et al. suggested that a pre-existing mental health condition may be associated with vulnerabilities in executive and cognitive function revealed by SARS-CoV-2 infection [31]. Prior history of mental illness may mitigate the psychological burden of long-term symptomatology of COVID-19 [32]. Currently, no agreement exists on a label for post-COVID-19 cognitive/mental disorders. The time of symptom onset after acute infection and the duration is still debated. Memory and attention-executive complaints and deficits, together with fatigue, anxiety, and depression symptoms are constantly described, but the objective assessment of these symptoms is still not standardized [33]. Lastly, because of all the limitations in our research, we are unable to conclude that the results given are applicable for all populations.

## 5. Conclusions

The long-term sequelae of COVID-19 is now considered to be a public health issue with great challenges for the community and health economic system. Multiple studies have confirmed the long-lasting multisystemic effects of COVID-19 [30]. Despite primary guidance from different studies, there is not yet a standardized approach to identify clinical manifestations and risk factors of post-COVID-19 symptoms. This issue is further complicated by the lack of a widely accepted definition, timeline, classification and diagnostic criteria, and treatment recommendations. What the global medical community needs at this time is a consensus definition of this long COVID syndrome [34]. Clinical characteristics of the disease will help in appropriate diagnosis, care, public health interventions and policy, and resource planning. Empirical data on the scale and scope of the problem are urgently needed to support the development of appropriate public health responses [15,35].

## Figures and Tables

**Figure 1 ijerph-20-06678-f001:**
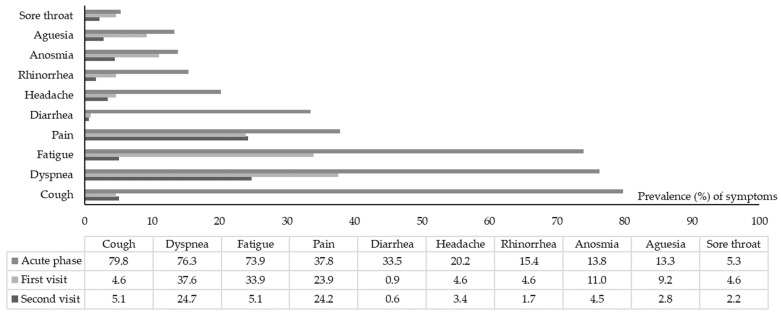
Prevalence of clinical symptoms at the acute phase, visit #1 and visit #2 in included patients.

**Table 1 ijerph-20-06678-t001:** Characteristics of included patients in a prospective study in 4 French university hospitals.

Variables	*N* = 189 (%)
Gender	
**Female**	76 (40.2)
Male	113 (59.8)
Age, mean (min–max)	63.4 (25–93)
BMI (kg/m^2^), mean (min–max), *n* = 152	27.4 (16.4–44.4)
**Past medical history**	151 (79.9)
Type of underlying diseases	
Cardiac failure	24 (12.7)
Cardiovascular diseases	84 (44.4)
Diabetes mellitus	36 (19)
Hypertension	68 (36)
Hypothyroidism	17 (9)
Immunodeficiency	10 (5.3)
Liver diseases	16 (8.5)
Lung diseases	12 (6.3)
Malignancy	21 (11.1)
Neurological diseases	9 (4.8)
Renal diseases	18 (9.5)
Rheumatic diseases	20 (10.6)
Other	100 (52.9)
**COVID-19 history**	
Delay (days) between onset of symptoms and hospital admission, mean (min–max)	8.2 (0–31)
**Symptoms at hospital admission, *N* (%)**	
Ageusia	25 (13.3)
Anosmia	26 (13.8)
Cough	150 (79.8)
Diarrhea	63 (33.5)
Fatigue	139 (73.9)
Headache	38 (20.2)
Irritability	6 (3.2)
Nausea	23 (12.2)
Pain	71 (37.8)
Pain location	
Abdominal	14 (19.7)
Chest	10 (14.1)
Joints	6 (8.5)
Muscular	52 (73.2)
Shortness of breath	124 (66)
Sore throat	10 (5.3)
Rhinorrhea	29 (15.4)
Other symptoms	48 (25.5)
Number of symptoms, mean (min–max)	5.0 (1–9)
Duration (days) of symptoms in symptomatic patients, mean (min–max) *n* = 164	24.5 (2–126)
**Signs at hospital admission, *N* (%)**	
Abnormal lung auscultation, *n* = 173	157 (83.1)
Abnormal lung X-ray when realized, *n* = 162	157 (96.9)
Coma, *n* = 173	1 (0.6)
Dyspnea, *n* = 173	132 (76.3)
Fever	171 (91)
Temperature (°C), mean (min–max)	37.8 (35.5–40)
**Biological data (Reference Range)**	
Complete Blood Count	
WBC (4–10 G/L), mean (min–max), *n* = 178	6.9 (1.5–22.8)
Neutrophils (1.8–7.5 G/L), mean (min–max), *n* = 178	5.3 (1.1–17.8)
Lymphocytes (1–4 G/L), mean (min–max), *n* = 178	1.1 (0.1–0.68)
Monocytes (0.2–0.9 G/L), mean (min–max), *n* = 178	0.5 (0.0–1.6)
Platelets (150–400 G/L), mean (min–max), *n* = 177	216.7 (71–618)
RBC (4–6 G/L), mean (min–max), *n* = 178	4.7 (2.8–6.5)
Hemoglobin (120–170 G/L), mean (min–max), *n* = 178	138.1 (70–199)
Biochemistry tests	
ALT (13–61 U/L), mean (min–max), *n* = 148	45.1 7–375)
AST (15–37 U/L), mean (min–max), *n* = 148	58.4 (12–671)
Creatinine (45–104 µmol/L), mean (min–max), *n* = 178	87.4 (30–669)
C-Reactive Protein (< 5 mg/L), mean (min–max), *n* = 167	86.0 (0–367.6)
LDH (87–241 U/L), mean (min–max), *n* = 62	362.2 (141–825)
Potassium (3.5–5.1 mmol/L), mean (min–max), *n* = 177	4.1 (2.8–6.2)
Prothrombin time (70–150 %), mean (min–max), *n* = 142	81.7 (18–126)
Sodium (136–145 mmol/L), mean (min–max), *n* = 179	136.0 (125–146)
Urea (2.5–9.2 mmol/L), mean (min–max), *n* = 178	6.9 (2.10–47.7)
**Prognosis**	
Admission to ICU	64 (33.9)
Development of complications during hospital stay	
Respiratory	46 (83.6)
Bacterial pneumonia	16 (29.1)
Other infections	11 (20)
Cardiac	8 (14.5)
Other complications	34 (20.5)
Length of stay (days), mean (min–max)	19.9 (1–184)

ALT, alanine transaminase; AST, aspartate transaminase; BMI, body mass index; CRP, C-reactive protein; ICU, intensive care unit; LDH, lactate dehydrogenase; RBC, red blood cells; WBC, white blood cells.

**Table 2 ijerph-20-06678-t002:** Prevalence of persistent clinical symptoms at the 2 follow-up visits in included patients.

	First Visit	Second Visit
Variables	*N* = 189 (%)	*N* = 178 (%)
**Delay between COVID-19 and first visit (months), mean (min–max)**	8.8 (6–11)	12.1 (10–13)
**Patients with at least one persistent clinical symptom**	109 (57.7)	103 (57.9)
**Persistent clinical symptoms**		
Ageusia	10 (9.2)	5 (2.8)
Anosmia	12 (11)	8 (4.5)
Confusion	4 (3.7)	1 (0.6)
Cough	5 (4.6)	9 (5.1)
Diarrhea	1 (0.9)	1 (0.6)
Dyspnea	41 (37.6)	44 (24.7)
Fatigue	37 (33.9)	9 (5.1)
Headache	5 (4.6)	6 (3.4)
Nausea	0 (0)	2 (1.1)
Pain	26 (23.9)	43 (24.2)
Pain localization		20 (11.2)
Abdominal	1 (3.8)	2 (10)
Articular	6 (23.1)	11 (55)
Muscular	15 (57.7)	11 (55)
Respiratory	6 (23.1)	4 (20)
Rhinorrhea	5 (4.6)	3 (1.7)
Sore throat	5 (4.6)	4 (2.2)
Other symptoms	54 (49.5)	50 (28.1)
**Readmission COVID-19 related**	4 (2.1)	1 (0.6)
**New clinical symptoms reported**	27 (14.3)	9 (5.1)
**Type of new reported clinical symptoms**		
Cough	0 (0)	1 (0.6)
Diarrhea	4 (14.8)	0 (0)
Dyspnea	5 (18.5)	3 (1.7)
Fatigue	6 (22.2)	5 (2.8)
Fever	5 (18.5)	1 (0.6)
Headache	5 (18.5)	2 (1.1)
Nausea	1 (3.7)	0 (0)
Pain	6 (22.2)	0 (0)
Rhinorrhea	2 (7.4)	0 (0)
Sore throat	3 (11.1)	0 (0)

**Table 3 ijerph-20-06678-t003:** Factors associated with persistent clinical symptom(s) at first and second follow-up visits in included patients’ results of univariate analysis.

	First Visit		Second Visit	
Variables	Crude OR, (IC95%)	*p*	Crude OR, (IC95%)	*p*
Gender (male)	0.82 (0.46–1.48)	0.52	0.73 (0.45–1.75)	0.73
Age, years	0.99 (0.97–1.01)	0.36	0.99 (0.97–1.02)	0.53
BMI > 30 kg/m^2^	2.91 (1.21–7.0)	0.02	2.83 (1.03–7.74)	0.04
At least one underlying disease	0.78 (0.38–1.59)	0.49	1.02 (0.44–2.32)	0.97
Cardiac failure	0.70 (0.30–1.65)	0.42	0.67 (0.24–1.89)	0.45
Cardiovascular diseases	1.05 (0.59–1.88)	0.87	1.13 (0.59–2.21)	0.72
Chronic pulmonary diseases	1.03 (0.32–3.37)	0.96	1.05 (0.28–3.89)	0.94
Diabetes mellitus	0.68 (0.33–1.41)	0.30	0.79 (0.33–1.91)	0.61
Immunodeficiency	0.72 (0.20–2.58)	0.62	0.68 (0.19–2.46)	0.56
Hypertension	0.98 (0.54–1.79)	0.95	1.02 (0.52–2.03)	0.95
Hypothyroidism	1.86 (0.63–5.50)	0.26	3.39 (0.71–16.29)	0.13
Liver diseases	1.68 (0.56–5.05)	0.35	1.64 (0.48–5.58)	0.43
Malignancy	1.22 (0.48–3.10)	0.68	0.86 (0.32–2.31)	0.76
Neurological diseases	0.35 (0.09–1.44)	0.15	0.21 (0.04–1.10)	0.07
Renal diseases	0.71 (0.27–1.88)	0.49	0.78 (0.27–2.27)	0.64
Rheumatic diseases	1.41 (0.54–3.72)	0.49	2.27 (0.70–7.42)	0.18
Delay (days) between onset of symptoms and hospital admission	1.04 (0.98–1.11)	0.22	1.06 (0.98–1.14)	0.15
Admission to ICU	2.03 (1.08–3.83)	0.03	2.48 (1.19–5.17)	0.02
Complications during hospital stay	2.51 (1.27–5.0)	0.008	2.51 (1.14–5.52)	0.02
Length of hospital stay (days)	1.01 (1.0–1.03)	0.05	1.01 (1.0–1.03)	0.11
Duration of initial symptoms (days)	1.02 (1.0–1.04)	0.04	1.03 (1.0–1.05)	0.04
Number of initial symptoms	1.12 (0.94–1.33)	0.20	1.21 (0.98–1.48)	0.08
Abnormal lung osculation	1.16 (0.41–3.28)	0.78	1.41 (0.41–4.89)	0.56
Ageusia	0.83 (0.39–2.13)	0.83	0.81 (0.34–2.02)	0.65
Anorexia	0.84 (0.36–1.92)	0.67	0.53 (0.21–1.31)	0.17
Anosmia	2.17 (0.87–5.45)	0.10	2.22 (0.82–6.01)	0.12
Cough	1.15 (0.56–2.35)	0.70	1.44 (0.64–3.26)	0.38
Diarrhea	0.86 (0.47–1.59)	0.63	0.97 (0.49–1.93)	0.94
Dyspnea	0.83 (0.41–1.69)	0.61	1.13 (0.49–2.59)	0.78
Fatigue	1.05 (0.54–2.02)	0.89	1.02 (0.49–2.13)	0.96
Fever	2.78 (0.98–7.86)	0.05	15 (1.85–121.88)	0.01
Headache	1.0 (0.48–2.05)	0.99	1.26 (0.55–2.89)	0.58
Irritability	0.72 (0.14–3.65)	0.69	0.45 (0.07–2.79)	0.39
Nausea	1.15 (0.47–2.80)	0.76	1.05 (0.40–2.76)	0.92
Pain	1.30 (0.71–2.38)	0.39	1.58 (0.80–3.14)	0.19
Rhinorrhea	0.87 (0.39–1.94)	0.74	1.16 (0.45–2.99)	0.76
Short of breath	1.81 (0.98–3.33)	0.06	2.05 (1.0–4.22)	0.05
Sore throat	0.71 (0.20–2.55)	0.60	0.93 (0.20–4.30)	0.90
Other symptoms	1.20 (0.52–2.81)	0.67	1.23 (0.45–3.33)	0.67
Laboratory data				
ALT (≥55 U/L)	1.15 (0.52–2.54)	0.74	1.34 (0.56–3.27)	0.52
AST according to quartiles (ref < 31)				0.10
31–42	3.49 (1.33–9.12)	0.01	2.95 (1.03–8.45)	0.04
>42	2.16 (0.95–4.88)		2.32 (0.91–5.88)	0.08
Creatinine (>104 µmol/L)	0.71 (0.3–1.68)	0.44	0.74 (0.29–1.88)	0.52
CRP (mg/L)	0.76 (0.21–2.70)	0.67	0.68 (0.16–2.83)	0.59
LDH (>400 U/L)	0.86 (0.30–2.47)	0.78	1.26 (0.36–4.42)	0.72
Potassium (>4.5 mmol/L)	1.00 (0.41–2.42)	1.0	1.06 (0.38–2.96)	0.92
Prothrombin (<70%)	0.92 (0.38–2.19)	0.85	0.53 (0.20–1.40)	0.20
Urea (>9.2 mmol/L)	0.92 (0.40–2.10)	0.84	1.02 (0.40–2.58)	0.97
Monocytes (>0.9 G/L)	1.13 (0.38–3.32)	0.83	1.06 (0.29–3.94)	0.93
Neutrophils (>7.5G/L)	1.29 (0.53–3.14)	0.57	1.41 (0.52–3.79)	0.50
WBC (>10 G/L)	1.35 (0.54–3.41)	0.52	1.48 (0.52–4.20)	0.46

ALT, alanine transaminase; AST, aspartate transaminase; BMI, body mass index; CI, confidence interval; CRP, C-reactive protein; ICU, intensive care unit; LDH, lactate dehydrogenase; OR, odds ratio; RBC, red blood cells; WBC, white blood cells.

**Table 4 ijerph-20-06678-t004:** Factors independently associated with persistent clinical symptom(s) at first and second follow-up visits in included patients’ results of the multivariate logistic regression.

	First Visit		Second Visit	
Variables	Adjusted OR (95% CI)	*p*	Adjusted OR (95% CI)	*p*
**Personal history**				
BMI > 30 kg/m^2^	3.52 (1.25–9.91)	0.02	3.80 (0.97–14.91)	0.06
Rheumatic diseases	-	-	5.25 (0.75–36.84)	0.09
**Factors related to acute COVID-19**				
Ageusia	-	-	0.17 (0.03–0.92)	0.04
Anosmia	-	-	13.34 (2.07–86.21)	0.006
AST according to quartiles (ref < 31)		0.003		0.03
31–42	8.68 (2.41–31.28)	0.001	10.27 (1.88–56.29)	0.007
>42	3.69 (1.32–10.31)	0.01	2.52 (0.60–10.68)	0.21
Delay (days) between onset of clinical symptoms and hospital admission	-	-	1.14 (1.01–1.29)	0.04
Pain	-	-	4.31 (1.23–15.05)	0.02
ICU stay	-	-	5.43 (1.39–21.25)	0.02

AST, aspartate transaminase; BMI, body mass index; CI, confidence interval; ICU, intensive care unit; OR: odds ratio.

## Data Availability

Data can be available on specific demand to be addressed to the corresponding author.

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
