# Peer review of "Factors Associated with Long COVID-19 in a French Multicentric Prospective Cohort Study"

_ijerph, 2023, doi:10.3390/ijerph20176678_

Round 1
Reviewer 1 Report
Overall, the researcher presented clearly
Here are some additional suggestions :
Abstrat : Method part should be present age of participants
Materials and Methods :
1.what is research design in this study, should be present it
2.Why this study presented A-value of <:20 in line 105 , should explain it
Results : Table 3 , 4 should be present p-value
Conclussion: should be add suggestions for appication of this research results ,and futher research
Author Response
Overall, the researcher presented clearly.
We would like to thank you for the attention you paid to this work.
Here are some additional suggestions :
Abstrat : Method part should be present age of participants
We agree with you and the information was added in method and results sections.
Materials and Methods :
1.what is research design in this study, should be present it
Thank you for this remark and the text was modified.
2.Why this study presented A-value of <:20 in line 105 , should explain it
Different models of multivariate logistic regression analysis were done to identify factors associated with COVID long (at each visit of follow-up). For each model, a stepwise forward selection was performed. P-values of ≤0.20 in univariate analysis was considered as the criteria for model inclusion.
We used a stepwise regression step-by-step iterative construction of a regression model that involves the selection of independent variables to be used in a final model. It involves adding or removing potential explanatory variables in succession and testing for statistical significance after each iteration. In our analyses the stepwise logistic regression was designed to find the most parsimonious set of predictors that are most effective in predicting the dependent variable (long covid: yes or no according to our definition).Variables are added to the logistic regression equation one at a time, using the statistical criterion of reducing the Likelihood error for the included variables. After each variable is entered, each of the included variables are tested to see if the model would be better off the variable were excluded. This part was completed in the revised version.
Results : Table 3 , 4 should be present p-value
The P values are already presented in the initial version (the column after those of Odds Ratio)
Conclussion: should be add suggestions for appication of this research results ,and futher research
Some points were added in the revised version.
Reviewer 2 Report
In this cohort study, Khanafer et al aim to explore the potential clinical factors that are associated with the prevalence of prolonged COVID symptoms after hospital discharge. Previously hospitalized patients were followed up for one year and interviewed every six months. They found, among multiple previously defined symptoms, Dyspnea, and pain. Using Multivariate logistic regression methodology, several factors, including Anosmia and Pain during initial infection, have been found to be associated with a higher risk of long COVID while, interestingly, an opposite association with ageusia is also revealed. The present study is solid, and the results are well-presented. Addressing of below comments would contribute to the clarity of the manuscript.
1. Please outline the aim/goal of the study in the introduction, it is not clear.
2. It might be helpful to re-order the symptoms in Figure 1 by their frequencies.
3. It seems to me that fever also showed to have a significant or at least trend toward significant association with the long COVID. Could you please discuss this?
4. It is worth discussing the future directions or what needs to be done as a follow-up of the present study.
Some grammar corrections are needed, i.e. lines 167-170.
Author Response
In this cohort study, Khanafer et al aim to explore the potential clinical factors that are associated with the prevalence of prolonged COVID symptoms after hospital discharge. Previously hospitalized patients were followed up for one year and interviewed every six months. They found, among multiple previously defined symptoms, Dyspnea, and pain. Using Multivariate logistic regression methodology, several factors, including Anosmia and Pain during initial infection, have been found to be associated with a higher risk of long COVID while, interestingly, an opposite association with ageusia is also revealed. The present study is solid, and the results are well-presented. Addressing of below comments would contribute to the clarity of the manuscript.
We would like to thank you for the attention you paid to this work.
- Please outline the aim/goal of the study in the introduction, it is not clear.
Thank you for this remark and the text was modified.
- It might be helpful to re-order the symptoms in Figure 1 by their frequencies.
We agree with you. In the revised version, the figure was modified according to your suggestions
- It seems to me that fever also showed to have a significant or at least trend toward significant association with the long COVID. Could you please discuss this?
Fever seems associated with long COVID at first visit (P=0.05 without reaching statistical significance) and the second follow-up visit (P=0.01). However, this association was not observed in multivariate analysis after adjustment on other variables.
In a recent publication, Belon et al. reported that mild fever (after Acute infections, including those due to Coronaviridae and other viruses) appears to improve outcome; it appears to diminish viral replication by several mechanisms, including virion entry into host cells and genome transcription, and improving host defence mechanisms against the pathogen. However, a fever may also damage host cellular and tissue function and increase metabolic demands. At temperatures at the lower end of the febrile range, the benefit of the fever appears to outweigh the detrimental effects. However, at higher temperatures, the outcome worsens, suggesting that the disadvantages of fever on the host predominate (doi: 10.12998/wjcc.v9.i2.296).
- It is worth discussing the future directions or what needs to be done as a follow-up of the present study.
Some modifications were done in the revised version.
Comments on the Quality of English Language
Some grammar corrections are needed, i.e. lines 167-170.
The typing error was corrected.
Reviewer 3 Report
Thank you for the opportunity to review this manuscript. In this paper, the authors enrolled volunteers from a pre-existing cohort of hospitalized patients with COVID-19 in France to assess the prevalence of symptoms associated with long COVID. Clearly, a great deal of effort and time was expended by the authors. My main methodologic concerns are the absence of a comparator group, which is important when assessing the significance of common symptoms such as fatigue and pain, and the large number of potential endpoints which increase the possibility of random findings.
Specific comments
Page 1, lines 40 and 41 – Spell out “COVID-19” on its first use (line 40), not its second use (line 41).
Page 2 – The introduction section summarizes some of the challenges with the definition of “long COVID”, but the introduction should also explain the purpose of the study and not abruptly stop and lead into the methods section.
Page 2, line 75 – Why did the authors exclude patients with multiple hospitalizations? I suspect it was to reduce confounding issues (such as disability related to other, non-COVID illnesses), but this should be explained in the methodology.
Page 2, line 78 – How did the participants from the current study differ from the overall initial cohort population? Were they different in age, sex, ethnicity, burden of comorbidities, etc.? It seems that 189 patients is a reasonable response rate, but we need to know a little more about their relationship to the original NOSO-COR cohort in order to assess the generalizability of these findings.
Page 2, line 86 – This should say “Data regarding the initial episode” or “data regarding initial episodes”, not “data regarding initial episode.”
Page 2, line 89 – This should say “the presence of at least one persisting clinical symptom”, not “symptoms.”
Page 2, line 90 – Given the close relationship between many symptoms of long COVID and cognitive/psychological symptoms, it seems odd to deliberately exclude those disorders. This requires explanation.
Page 2, line 92 – This definition of long COVID seems very broad, so broad that it seems to require some sort of control group for comparison. If all it takes to have “long COVID” is to have a single symptom of some sort within 6-12 months after a primary COVID-19 infection, then the entire world probably has long COVID.
Page 3, line 117 – Just to reiterate that comparing this group to the original complete cohort would be useful.
Page 3, line 127 – When were these laboratory tests obtained? At the time of the study visits, or during the index hospitalization?
Page 3-4, table 1 – The “number of underlying diseases” is difficult to assess in a standardized way. Consider rendering this in some manner of standardized format, e.g., the Charlson or Elixhauser comorbidity indices.
Page 6, table 2 – I would encourage standardization of your terminology here. I am unsure how fatigue is different from asthenia, for example, but they are listed as separate terms in the table.
Page 7, line 167 – The sentence beginning with “While patients that had suffered from ageusia…” is incomplete. I think this was meant to follow the preceding sentence as “…with persistent symptoms 12 months after acute COVID-19, while patients who had suffered from ageusia…”
Page 7, line 170 – This should say “…of long COVID, but the association did not reach statistical significance.”
Page 7, table 3 – The authors’ finding that anosmia was associated with a higher risk of long COVID while ageusia was protective seems to lack biological plausibility. As I look at table 3, the very large number of measured variables suggests that this is most likely a coincidental finding. If you have over 50 possible endpoints, the odds that one of them will be randomly “significant” is very high.
Page 9, line 188 – It would be useful to have some idea about the prevalence of these (very common) symptoms in the general population to assess the impact of long COVID.
Page 9, line 243 – How exactly does a face-to-face interview with a medical doctor reduce information bias?
The English grammar is clearly better than my French, but there are numerous grammatical and punctuation errors; I have pointed out a few of them above, but the paper in its current form does not meet the standards of a scientific publication from a standpoint of language quality.
Author Response
Thank you for the opportunity to review this manuscript. In this paper, the authors enrolled volunteers from a pre-existing cohort of hospitalized patients with COVID-19 in France to assess the prevalence of symptoms associated with long COVID. Clearly, a great deal of effort and time was expended by the authors. My main methodologic concerns are the absence of a comparator group, which is important when assessing the significance of common symptoms such as fatigue and pain, and the large number of potential endpoints which increase the possibility of random findings.
We would like to thank you for the attention you paid to this work.
Specific comments
Page 1, lines 40 and 41 – Spell out “COVID-19” on its first use (line 40), not its second use (line 41).
The typing error was corrected.
Page 2 – The introduction section summarizes some of the challenges with the definition of “long COVID”, but the introduction should also explain the purpose of the study and not abruptly stop and lead into the methods section.
Thank you for this remark and the text was modified.
Page 2, line 75 – Why did the authors exclude patients with multiple hospitalizations? I suspect it was to reduce confounding issues (such as disability related to other, non-COVID illnesses), but this should be explained in the methodology.
This comment was considered in the revised version.
Page 2, line 78 – How did the participants from the current study differ from the overall initial cohort population? Were they different in age, sex, ethnicity, burden of comorbidities, etc.? It seems that 189 patients is a reasonable response rate, but we need to know a little more about their relationship to the original NOSO-COR cohort in order to assess the generalizability of these findings.
There were no significant differences regrding age and gender between the two groups. Please notice that, the participation was based on volontariat without any selection for included patients.
Page 2, line 86 – This should say “Data regarding the initial episode” or “data regarding initial episodes”, not “data regarding initial episode.”
The typing error was corrected.
Page 2, line 89 – This should say “the presence of at least one persisting clinical symptom”, not “symptoms.”
The typing error was corrected.
Page 2, line 90 – Given the close relationship between many symptoms of long COVID and cognitive/psychological symptoms, it seems odd to deliberately exclude those disorders. This requires explanation.
We agree with you about the probable relathioship between cognitive/psychological symptoms and long COVID. Data on cognitive/psychological symptoms were not collected in this study. This information was discussed in the revised version.
Page 2, line 92 – This definition of long COVID seems very broad, so broad that it seems to require some sort of control group for comparison. If all it takes to have “long COVID” is to have a single symptom of some sort within 6-12 months after a primary COVID-19 infection, then the entire world probably has long COVID.
The study population is limited to laboratory-confirmed COVID-19 patients hospitalized in 4 university-affiliated hospitals (Hospices Civils de Lyon, Lyon, France), who have been previously enrolled in the NOSO-COR cohort study [11]. The objectives were to estimate the prevalence of persistent symptoms and/or the onset of new one after an initial epiosde of COVID-19 and in a second time to identify factors associated with lon COVID among infected people. Therefore the control group for the study is the patients with no “long COVID” at follow-up visits.
Page 3, line 117 – Just to reiterate that comparing this group to the original complete cohort would be useful.
The information was added as mentionned before.
Page 3, line 127 – When were these laboratory tests obtained? At the time of the study visits, or during the index hospitalization?
Laboratory tests were obtained during index hospitalization and follow-up visits
Page 3-4, table 1 – The “number of underlying diseases” is difficult to assess in a standardized way. Consider rendering this in some manner of standardized format, e.g., the Charlson or Elixhauser comorbidity indices.
We agree with you, this variable is not so relevant and was deleted in the revised version.
Page 6, table 2 – I would encourage standardization of your terminology here. I am unsure how fatigue is different from asthenia, for example, but they are listed as separate terms in the table.
The typing error was corrected.
Page 7, line 167 – The sentence beginning with “While patients that had suffered from ageusia…” is incomplete. I think this was meant to follow the preceding sentence as “…with persistent symptoms 12 months after acute COVID-19, while patients who had suffered from ageusia…”
The typing error was corrected.
Page 7, line 170 – This should say “…of long COVID, but the association did not reach statistical significance.”
The typing error was corrected.
Page 7, table 3 – The authors’ finding that anosmia was associated with a higher risk of long COVID while ageusia was protective seems to lack biological plausibility. As I look at table 3, the very large number of measured variables suggests that this is most likely a coincidental finding. If you have over 50 possible endpoints, the odds that one of them will be randomly “significant” is very high.
Different models of multivariate logistic regression analysis were done to identify factors associated with COVID long (at each visit of follow-up). For each model, a stepwise forward selection was performed. We used a stepwise regression which is a step-by-step iterative construction of a regression model that involves the selection of independent variables to be used in a final model. It involves adding or removing potential explanatory variables in succession and testing for statistical significance after each iteration. In our analyses the stepwise logistic regression was designed to find the most parsimonious set of predictors that are most effective in predicting the dependent variable (long covid: yes or no according to our definition).Variables are added to the logistic regression equation one at a time, using the statistical criterion of reducing the Likelihood error for the included variables. After each variable is entered, each of the included variables are tested to see if the model would be better off the variable were excluded. This part was completed in the revised version.
Page 9, line 188 – It would be useful to have some idea about the prevalence of these (very common) symptoms in the general population to assess the impact of long COVID.
Our study was conducted in patients hospitalized for their COVID-19 so the comparaison with general population is not possible. Moreover how we can exclude a relathioship between the prevalence of symptoms in general population and other illnesses? Symptoms were collected during the index hospitalization in a standardized way and similary during 2 follow-up visits which significantly different from data reported in general population in some countries.
Page 9, line 243 – How exactly does a face-to-face interview with a medical doctor reduce information bias?
Information bias is more frequent when data were obtained via a self-administred questionnaire, a medical recod or phone-interview especially if it is conducted with people from different backgrounds.
The interview was not limited to complete a questionnaire. In fact, patients were invited to two ambulatory follow-ups medical visits 6-8 months and one year after the initial COVID-19 onset. After a medical exam, the clinician completed a standardized questionnaire to ensure consistency in data collection.
Comments on the Quality of English Language
The English grammar is clearly better than my French, but there are numerous grammatical and punctuation errors; I have pointed out a few of them above, but the paper in its current form does not meet the standards of a scientific publication from a standpoint of language quality.
Your remark was considered for the revised version.
Round 2
Reviewer 3 Report
Thank you for the opportunity to review this revised manuscript. The authors have made improvements from the original draft that are reasonable. I still have some difficulties with the description of this study, given the absence of a proper definition for "long COVID". (This study is not unique in this regard, however.) As a cross-sectional epidemiological study of long-term symptom persistence in patients who have been hospitalized with COVID-19, it is a reasonable paper. However, the frequency of these symptoms in the general, non-hospitalized population makes the data challenging to interpret. Despite this, it is acceptably well-written.
Significantly improved. There are some minor errors (e.g., page 8, line 205 - should say "reach statistical significance", not "rich statistical significance), but I defer them to the copy editors for correction.
Author Response
Dear reviewer,
We would like to thank you for the attention you paid to this work. As mentionned in the response letter, our study was conducted in patients hospitalized for their COVID-19 so the comparaison with general population was not possible. Moreover how we can exclude a relathioship between the prevalence of symptoms in general population and other illnesses? Symptoms were collected during the index hospitalization in a standardized way and similary during 2 follow-up visits which significantly different from data reported in general population in some countries.
The sentence regarding the BMI was modified: the association was not significant in multivariate analysis, OR: 3.80 95%CI: 0.97-14.91.
Some elements, highlighted in green, were added in the new revised revision.